# Unlocking insights from complex data: Leveraging heat maps for decision-making in LMIC

**Muhammad Ibrahim[1], Olan Naz[1], Amal Fatima Mohiuddin[1], Habib Ur Rehman[1], Adnan Ahmad Khan[1,2]***

**1** Research and Development Solutions (RADS), Islamabad, Pakistan, **2** Ministry of National Health Services, Regulations and Coordination (MoNHSRC), Islamabad, Pakistan

* adnan@resdev.org

## Abstract

### Introduction

In low- and middle-income countries (LMICs), health outcomes are often constrained by inadequate or misdirected resource allocation and limited access to services such as contraception and immunization. We explore the use of spatial heat maps to analyze the stock availability and dispensation/vaccination patterns of contraceptives and vaccines in Pakistan.

### Methods

We used data from national contraceptive (cLMIS) and vaccine logistics management information systems (vLMIS). We applied univariate, bivariate, and trivariate spatial heat maps to assess contraceptive and vaccine stock levels, dispensation/ vaccination, and wastage across districts. For contraception, we standardized stocks per 100,000 married women of reproductive age (MWRA) and dispensation rates. In immunization, we focused on Pentavalent-3 (Penta3) vaccine outreach, dropout rates, and Bacillus Calmette–Guérin (BCG) vaccine wastage.

### Results

Temporal and spatial variations highlighted regional disparities, revealing that developed regions like Punjab had better stock availability, while underserved areas like Balochistan faced higher dispensation rates and stockouts. We also show the effect of inputs (supplies, outreach) on dispensation and utilization of contraceptives and vaccines, respectively. Finally, we depict how these visualizations can help track changes in programming over time.

**Data availability statement:** The data has been acquired from "The Chemonics, Pakistan" under collaboration with Research and Development Solutions (RADS), Islamabad, Pakistan. However, the data will be available upon request as Chemonics collects data from public and private stakeholders and they provide data on special request. Data can be requested from following email addresses: support@lmis.gov.pk; muhahmed@chemonics.com; mtariq@chemonics.com Shapefiles data can be easily accessed through online link: https://www.geoboundaries.org/ The website has Creative Commons Attribution International (CC BY) and Public Domain/No Restrictions under which we are free to use the material for any purpose.

**Funding:** This work has been funded by the Bill & Melinda Gates Foundation (INV 051108). The funding agency had no role in the development of this manuscript.

**Competing interests:** The authors have declared that no competing interests exist.

## Conclusions

Our findings show that integrating spatial data visualization with health logistics data identifies critical gaps in health service supply and demand, guiding policymakers in resource allocation, stock management, and service outreach. This scalable approach suits systems with limited analytical resources, as many analyses can be automated and embedded in datasets, providing policymakers with a focused set of visualizations for interpretation, avoiding the need for extensive training or deploying analysis teams at local levels. By leveraging spatial and temporal data, this method supports efficient health system strengthening and resource allocation in LMIC.

## Introduction

Access to effective services is a major hurdle to family planning (FP) services in Pakistan and accounts for over 70% of non-use by those wishing to use FP [1]. A key part of access to effective contraception is the availability of contraceptives at facilities [2]. Similarly, vaccine program implementation in LMICs face multifaceted challenges, including weak decision-making, poor logistics, and inadequate service delivery for preventable childhood illnesses [3]. Inability to ensure services and supplies in a timely manner leads to missed opportunities in family planning or other preventive or promotive services and exacerbate health outcomes and social disparities [4,5]. While policymakers seek to address these gaps, it is imperative that their decisions are guided by evidence—specifically, data on how services and supplies are managed, how they are utilized, and where critical gaps exist, thereby improving health and well-being.

While considerable data are available from procurement, monitoring and evaluation systems within healthcare, as well as from national-level community surveys, their potential for informed policymaking remains largely untapped [6]. This is due to limited capacity to consistently and timely harness, manage and interpret data [7]. Often, this interpretation would include triangulation of disparate data sources, analysis, and depiction into forms, such as visualization to make data more accessible for decision makers who may not be data experts [8–11]. Additionally, information such as population and program data change rapidly, and interpretations must keep pace with these changes to stay relevant. This underscores the need to utilize adaptable visualization methods that can be replicated routinely and frequently to facilitate real-time interpretations [10,12–14].

Basic bar graphs and pie charts are often the mainstay of presentations but convey limited information. By contrast, spatial heat maps offer greater flexibility and enhanced readability, effectively illustrating regional variations, highlighting hotspots or time variations [15]. One key advantage of spatial heat maps is that they consistently represent the surface area of a country, province, or district using geographic coordinates, ensuring accurate spatial context and allowing for clear comparisons across regions. In addition to depicting simplified indicators, such as stock levels,

total dispensation, or facility counts, these maps can also simplify more complex visualizations such as the intersection of multi-indicators and can be highlighted through interactive visualization [14].

Decision-making is a cognitive process where a person evaluates factors to solve a problem [16], depending on the complexity of the decision (such as the number of criteria and outcomes involved) [17]. While simpler decisions work with simple heuristics, complex ones require solutions that include ranking the quality and utility of data based on predetermined criteria [18]. Limited guidelines illustrate how to design indicators to support complex reasoning across different contexts [19]. Visualizations utilize the brain's sensory capacity to depict complex relationships between data points [20], such as simplifying the relationships of indicators not just across spatial dimensions, but also relate them over time and with other variables [21,22].

We demonstrate how spatial heat maps can be used as effective data visualization tools to support evidence-based decision-making in health programs within LMIC, particularly Pakistan, using data from the national contraceptive and vaccine logistics management information system (cLMIS and vLMIS) for 2022. The approach seeks to identify geographic disparities in supply and demand of commodities/vaccines, variations overtime and highlight inefficiencies. The objective is to illustrate how univariate, bivariate and trivariate heat maps can translate complex datasets into accessible, actionable insights for policymakers and district-level health managers, ultimately promoting more equitable and efficient resource allocation.

## Data and methodology

Data from the cLMIS (contraceptives) and vLMIS (vaccines) portals provided by The Chemonics, through their work with the USAID Global Health Supply Chain Program-Procurement and Supply Management project in Pakistan. Chemonics International is a global professional services firm focused on international development, working to find solutions for development in over 100 countries. They have a presence in Islamabad, Pakistan, and implement various projects related to development and supply chain. In alignment with Chemonics' reporting conventions, the term "consumption" in the cLMIS refers to the dispensation of contraceptive commodities from health facilities, not verified end-user utilization. To avoid misinterpretation, we refer to this variable as dispensation moving forward. Similarly, references to the demand side in this study pertain to the acquisition of commodities from facilities, and not to the actual use of contraceptives by clients.

### Data

Stocks and dispensation data for contraceptives were extracted, while vaccine data included indicators such as live birth targets (from now on referred as targeted population), stocks, vaccinations, and wastage. The supply chain for both contraceptives and vaccines are tracked by The Chemonics team from the central warehouse to provincial warehouses, district warehouses, and finally to individual facilities. Data were collected from a broad range of public service providers—such as the Pakistan Welfare Department (PWD) and the People's Primary Healthcare Initiative (PPHI)—as well as private organizations including Marie Stopes Society (MSS), DKT Pakistan, and Greenstar. While the dataset is comprehensive, it is subject to limitations such as delayed reporting and human error, which are discussed in the limitation section.

The focus was on the final stage of supply where stocks are bought by consumers. Data were aggregated at the district level to create spatial heat maps for broad discussion, which can be replicated at a granular level by district leadership. Monthly data for the calendar year 2022 were aggregated to provide annual estimates for both contraceptives and vaccines, except for one visual used to capture monthly disparities. All spatial heat maps are made using STATA 17.0 [23]. The administrative boundary shapefiles for Pakistan used in this study were obtained from the geoBoundaries Global Database, an open-access resource provided by the William & Mary geoLab [24]. These datasets are distributed under the Creative Commons Attribution 4.0 International License (CC BY 4.0), which permits unrestricted use, distribution, and reproduction in any medium, provided the original source is properly cited.

## Variables construction rationale

Contraceptives data, comprising only two main variables—stocks and dispensation—required careful assessment. "Stocks" refers to the quantity of commodities available at a health facility, while "dispensation" denotes the number of commodities dispensed to users from that facility. Simply displaying these variables on a spatial heat map is insufficient, as it fails to provide meaningful insights. Districts with higher populations naturally require more stock, and dispensation alone does not indicate whether the available stocks are being appropriately bought. For instance, if a district has higher stocks relative to other districts but a smaller population, and the dispensation is below 20%, the information can be misleading. A decision-maker might incorrectly conclude that Married Women of Reproductive Age (MWRA) are not buying enough contraceptives, whereas the issue could be overstocking. Therefore, it is essential to standardize these variables to ensure comparability across districts and enable a more accurate interpretation of the data. Analyzing both stock availability and dispensation indicators in tandem provides a more comprehensive understanding of supply chain performance and service utilization patterns.

## Data standardization

To standardize the stocks, the available stock per district was divided by the number of MWRA in a district and multiplied by 100,000. MWRA numbers were obtained from the Population Census 2017, resulting in the variable "Stocks available per 100,000 MWRA per district" formula shown in Table 1. The quartile distribution of these standardized values helps assess whether stock levels are adequate relative to the population in each district (further discussed in coming subsection).

After 2017, several new districts were added to the cLMIS portal which were not present in the population census data. Therefore, in the cLMIS data, we aggregated the values for the following districts: Sibi and Lehri, Chitral Lower and Chitral Upper, Kohistan Upper and Kohistan Lower, Duki and Loralai, Killa Abdullah and Chaman, and Shaheed Sikandar Abad and Kalat. Furthermore, cLMIS categorized Karachi as a single entity along with the seven districts within Karachi city, as a result, we aggregated all the data of Karachi city. These values were then categorized into five quantiles. Lastly, the shapefiles used in our analysis did not include separate administrative boundaries for Sujawal, which was encompassed within Thatta district, and Larkana, which was included in Kamber district. Therefore, data for these areas were aggregated accordingly within Sindh. Similarly, in Punjab, Chiniot was represented as part of Jhang, and Nankana Sahib was included within Sheikhupura; thus, values for these districts were merged within their respective parent districts.

Dispensation was standardized by calculating the percentage of available stock bought per district. The proportion of stocks bought serves as an indicator of demand for available commodities. For example, if dispensation falls below 25% or 50% of the available stock, it suggests underutilization and potential misalignment between supply and demand. We divided this indicator into four categories: up to 25% of stock dispensed (very poor dispensation pattern), 25–50% dispensed (relatively poor), 50–75% dispensed (relatively better), and above 75% dispensed (very better rate).

Separately, these spatial heat maps provide limited information, but when analyzed together, they offer more comprehensive insights for targeted policy development.

### Setting up bivariate spatial heat maps

Simple heat maps offer a significant improvement over basic tables, particularly when analyzing data across a large number of geographic units—such as Pakistan's 150 + districts. Initially, two separate spatial maps were generated to display stock availability and dispensation independently, highlighting how each indicator can be interpreted on its own. However, interpreting them side by side proved challenging. To address this and enhance clarity, a bivariate spatial heat map was developed to present both indicators simultaneously, allowing for a more integrated and intuitive understanding of supply and demand dynamics.

**Table 1. Indicators for Spatial Heat Maps per District.**

| Sr. No. | Indicator | Formula | Cut-off points of Data in Spatial Maps | Commodity/Vaccine {Illustrated in Figure No.} |
|---|---|---|---|---|
| | (1) | (2) | (3) | (4) |
| 1 | Stocks per 100,000 MWRA | $\left(\frac{Total\ Stocks}{MWRA}\right) \times 100,000$ | 0-41,000, 41,000-127,000, 127,000-297,000, >297,000 | Condoms {Figure 3} |
| | | | 0-4,900, 4,900−12,700, 12,700−19,700, >19,700 | Injections {Figure 3} |
| | | | 0-1,100, 1,100−3,400, 3,400−6,100, >6,100 | IUDs {Figure 3} |
| 2 | Dispensation Proportion of Stocks Available | $\left(\frac{Total\ Dispensation}{Total\ Stocks}\right) \times 100$ | 0-25%, 25-50%, 50-75%, 75-100% | Condoms, Injections & IUDs {Figure 2 & 3} |
| 3 | Pentavalent-3 outreach per facility | $\left(\frac{Vaccinations\ through\ outreach}{No.\ of\ Facilities}\right)$ | 0-25, 25-50, 50-75, >75 | Pentavalent 3 {Figure 4} |
| 4 | Pentavalent-3 Dropout Rates | $\left(\frac{Penta1-Penta3}{Penta1}\right) \times 100$ | <-50%, −50−0%, 0-50%, >50% | Pentavalent 3 {Figure 4} |
| 5 | Stocks per 1,000 targeted population | $\left(\frac{Total\ Stocks}{Live\ Birth\ Targets}\right) \times 1,000$ | 1,600-1,850, 1,850−2,000, 2,000-2,300, >2,300 | BCG {Figure 5} |
| 6 | Vaccinations per 1,000 targeted population | $\left(\frac{Total\ Vaccinations}{Live\ Birth\ Targets}\right) \times 1,000$ | 800-930, 930−1,060, 1,060−1,190, >1,190 | BCG {Figure 5} |
| 7 | Proportion of Wastage per Available Stock | $\left(\frac{Total\ Wastage}{Total\ Stocks\ Available}\right) \times 100$ | 0-100% | BCG {Figure 5} |

*Note: Above table doesn't show the cut off points for the univariate spatial maps created for contraceptives, as the values are visible on the legend. For contraceptives, Sr. No.1 to 3, cut offs are rounded off to nearest hundredth.*

Figure 1. Regional Disparities in Contraceptive Availability and Consumption in Pakistan.

On the y-axis, stocks per 100,000 MWRA are shown, now divided into four quartiles, with lighter shades indicating higher availability. On the x-axis, color variations represent changes in stock dispensation as previously discussed (up to 25%, 25–50%, and so on). This bivariate structure allows for the examination of sixteen distinct data points, as illustrated in results section (Fig 3).

The bivariate spatial heat maps for condoms, injections, and intrauterine devices (IUDs) are analyzed both separately and all three together. Condoms, being a one-time use contraceptive, show different stock and dispensation patterns compared to injections, which provide contraception for one to three months, and IUDs, which last five to eight years. Comparing these three bivariate spatial maps reveals varying preferences and usage patterns among MWRA (Fig 3).

With the utilization of bivariate spatial maps established for interconnected indicators, their application is extended to immunization indicators. The focus is on Sindh province, as the Chemonics team validated the data on their portal for this

region for vaccinations. There is a notable link between outreach vaccination services and vaccination dropout rates in Pakistan [25,26]. The Pentavalent vaccine third dose (Penta3) vaccine is crucial in protecting children against diphtheria, tetanus, pertussis (whooping cough), hepatitis B, and Haemophilus influenzae type b (Hib). Penta3 outreach vaccination outcomes are analyzed with Penta3 dropout rates. Calculations of indicators are shown in Table 1. A dropout rate refers to the percentage of children who received the first dose of the vaccine but did not complete the third dose. Monthly graphs were created to display Penta3 doses administered through outreach against Penta3 dropout rates. Demonstrating monthly bivariate maps reveal significant spatial and temporal variations pertinent to vaccination coverage.

## Spatial autocorrelation analysis

To assess spatial patterns in contraceptive indicators, we applied Local Indicators of Spatial Association (LISA) using Local Moran's I statistics for three methods: condoms, injectables, and IUDs. Spatial autocorrelation analysis was performed to identify statistically significant clustering or dispersion in the indicators across geographic locations. Only districts with complete data for each indicator were included; immunization indicators were excluded from this analysis because immunization section has fewer observations, we only have complete data of districts in Sindh province which gives us 27 data points.

This method quantifies the degree of spatial association for each district and identifies localized clusters (e.g., hotspots and cold spots) or spatial outliers [27]. Analyses were conducted using a first-order Queen contiguity spatial weights matrix, which defines neighboring districts as those sharing either a common boundary or a vertex, making it suitable for irregular administrative polygons. Spatial correlation analyses were conducted using GeoDa software [28]. The Local Moran's I analysis produced three complementary outputs:

- Local Moran's I statistic for each district, indicating the strength and direction of its spatial association with neighboring districts.

- LISA Cluster Map, classifying districts into four cluster types: high–high (hotspots), low–low (cold spots), high–low (low outliers), and low–high (high outliers)—based on indicator values relative to their neighbors.

- LISA Significance Map, identifying districts where the observed spatial association is statistically significant, based on 999 random permutations under the null hypothesis of spatial randomness ($\alpha = 0.05$).

In addition to univariate analyses, we conducted bivariate Local Moran's I to assess spatial relationships between stock levels and dispensation rates for the same contraceptive methods. Bivariate LISA examines whether high stock levels in a district are associated with high dispensation rates in neighboring districts, and vice versa, allowing exploration of cross-variable spatial dependencies and identifying potential spillover effects between adjacent administrative areas.

## Setting up trivariate spatial heat map

Lastly, unlike contraceptives, vaccine analysis requires the inclusion of an additional variable—wastage—to fully assess program performance. To accommodate this, trivariate spatial maps were developed, integrating stock availability, dispensation, and wastage to provide a more comprehensive evaluation of vaccine supply dynamics. Data was taken of Bacillus Calmette-Guerin (BCG) vaccine for 2022 across Sindh. Vaccine wastage is a crucial indicator that complements dispensation patterns [29], referred to as vaccinations. A new visualization strategy was employed to avoid confusion from using multiple colors. This approach integrates information from three variables in an easy-to-understand form. The base color on the spatial map represents stock availability per targeted population. Circles of varying size and color (red indicating low, green indicating adequate) depict vaccination rates per targeted population in each district. Additionally, pie charts overlaid on the circles represent wastage as a percentage of the available stock per district.

## Cut-off points for spatial heat maps

The classification thresholds used in the maps were primarily determined using the quartile method, which provides a non-parametric and distribution-independent approach to categorizing values. This method ensures that each category contains an equal number of observations, facilitating standardized comparisons across districts. It also mitigates the influence of outliers and avoids the arbitrariness associated with other classification schemes, such as equal intervals or natural breaks.

To assess the robustness of this approach, we conducted sensitivity analyses using an alternative equal-interval classification technique. While the equal-interval method divides the full range into equal-width classes, it obscures mid- and lower-range differences due to the skewed distribution of stock-to-population ratios, making cross-district comparisons less meaningful. In contrast, the quartile method provided a more balanced and comparable classification. Consequently, it was retained for its simplicity, interpretability, and policy relevance.

An exception to the quartile classification approach was applied to two indicators: (1) the proportion of available stock that was dispensed, and (2) Pentavalent-3 dropout rates. For these measures, predefined percentage-based thresholds were used instead of data-driven quartiles. This decision reflects the continuous nature of both indicators—ranging from 0% to 100% for stock dispensation, and from negative to positive percentages for dropout rates (e.g., less than –50% to more than +50%). These fixed intervals provide a more intuitive and policy-relevant interpretation of stock movement and service coverage. The specific cut-off points are detailed in Table 1.

## Results

Distribution and dispensation of condoms is used to illustrate the considerable regional variation across Pakistan. Fig 1 describes these in terms of absolute numbers. Central/ eastern regions, particularly Punjab, have the highest stock levels, sometimes exceeding 2.5 million condoms in some districts while south (parts of Sindh), west (Balochistan) and the north-west (Khyber Pakhtunkhwa – KP) have lower stock levels, ranging from 51–972 thousand condoms per district. Central and northeastern areas display moderate to high stocks, ranging from 0.9 to over 2.6 million. Dispensation patterns are similar. The same areas that have the lowest stocks also have the lowest dispensation, while those with the highest stocks have the most dispensation.

However, simple depictions of quantity does not account for the demand for such commodities. For this we adjusted the quantities for the population and depicted quantities per 100,000 MWRA (Fig 2). The resulting picture (Fig 2) remains similar for stocks in that, areas that had the least stocks overall also have the lowest stocks when adjected for population. However, the dispensation patterns change dramatically. Areas in southwest of Pakistan (province of Balochistan), south and southeast (province of Sindh) and northwest (province of KP) that had the least stocks also have the most of their stocks dispensed, resulting in stockouts. In the east (most of Punjab), where the more stock was supplied, relatively lesser proportion of the stock was dispensed, suggesting oversupply of condoms relative to the population-based need.

## Bivariate Spatial Maps and importance of its interpretation

This increasing level of complexity from matching supplies and dispensation can be depicted in a single figure to give decision makers one composite figure to understand the situation. Fig 3 shows the layering of stocks and dispensation together to show the interaction between them. The Y axis shows stocks per 100,000 MWRA, and the X axis shows percentage of dispensation against stock availability. Fig 2 confirms that districts in the northwest of Pakistan, and areas south and northeast of Balochistan that have the least supplies also have the least dispensation. While in central and eastern districts there are higher stocks and relatively low dispensation.

The pattern for contraceptive injections is very similar to condoms. However, the situation is very different for IUCD. There is the least stocking and dispensation in Balochistan (west). Sindh (south) sees moderate to high stocks but low dispensation while Punjab (central and east) has the highest supplies and low to moderate dispensation.

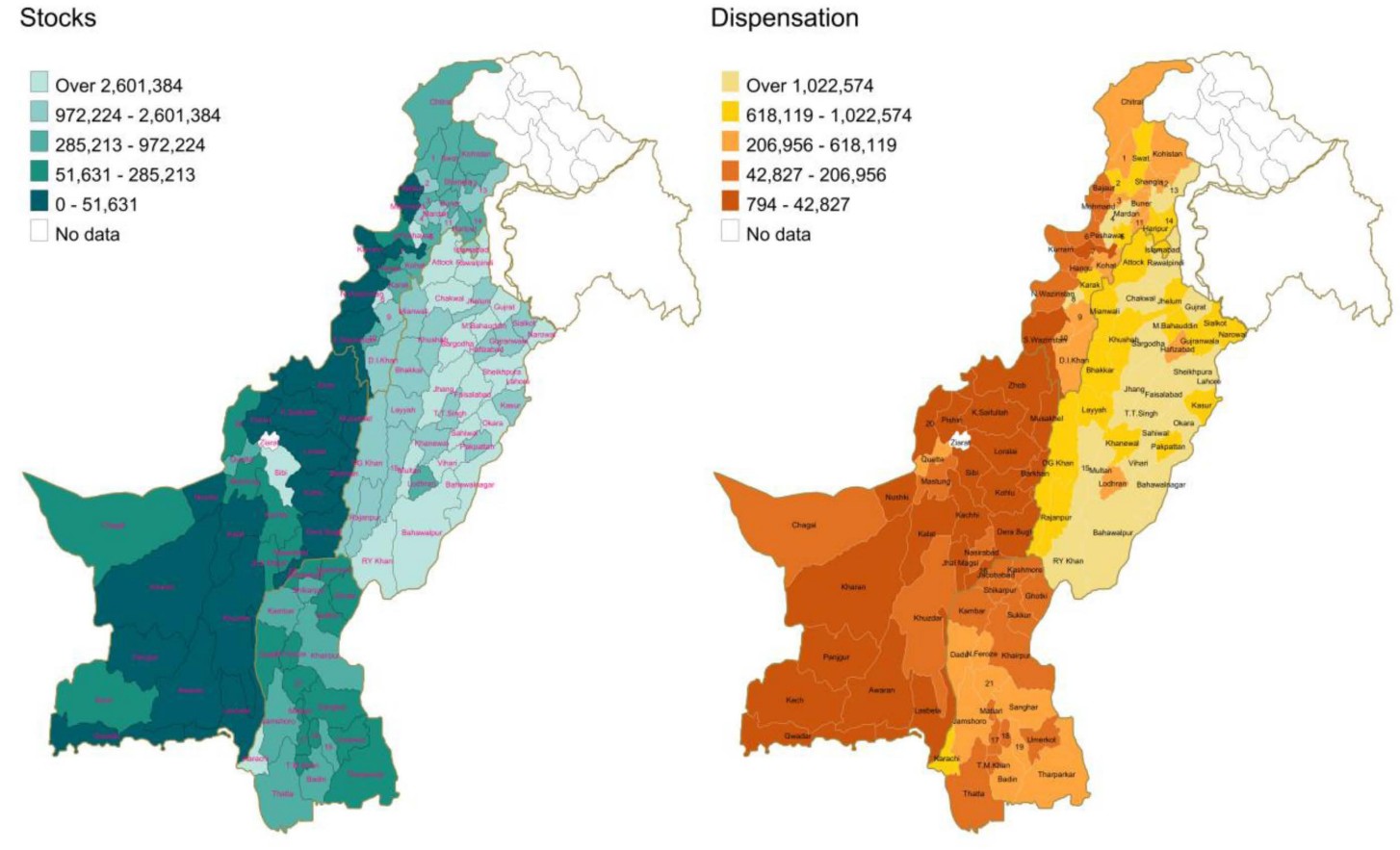

1.Upper Dir 2.Lower Dir 3.Malakand 4.Charsadda 5.Nowshera 6.Khyber 7.Orakzai 8.Bannu 9.Lakki Marwat 10.Tank 11.Swabi 12.Battagram 13.Mansehra
14.Abbottabad 15.Muzaffargarh 16.Jafarabad 17.Hyderabad 18.Tando Allahyar 19.Mirpurkhas 20.Killa Abdullah 21.Shaheed BenazirAbad

**Fig 1. Stocks and Dispensation of Condoms in Actual Quantities.** *Map Shape Files Source: Reprinted from* geoBoundaries *under a CC BY license, some modifications were made (included disputed area of Jammu & Kashmir) and used for illustrative purposes only.*

## Spatial autocorrelation of contraceptives

Local Moran's I statistics were calculated for contraceptive stocks and dispensation across districts, as well as for bivariate relationships between stocks and dispensation (Table 2). Positive Moran's I values for univariate stocks of condoms (0.179), injections (0.159), and IUDs (0.212) indicate modest positive spatial autocorrelation, suggesting clustering of similar values. Dispensation measures showed weaker positive autocorrelation, with Moran's I ranging from 0.038 (condoms) to 0.162 (injections), scatterplots are presented in S1 Appendix.

For condom stocks, Low–Low clusters were concentrated in southern Balochistan and most of Sindh (excluding Karachi, Tharparkar, and Umerkot), while High–High clusters appeared in a few KP districts. Dispensation Low–Low clusters were mainly in central and southern Punjab, with Dera Ismail Khan (KP) also included. Bivariate analysis showed that for dispensation vs stock of condoms, High–Low clusters (high dispensation, low neighboring stocks) occurred in nine districts of Balochistan and two of Sindh, while stock vs dispensation revealed several Low–High clusters, indicating low stocks surrounded by high-dispensation neighbors.

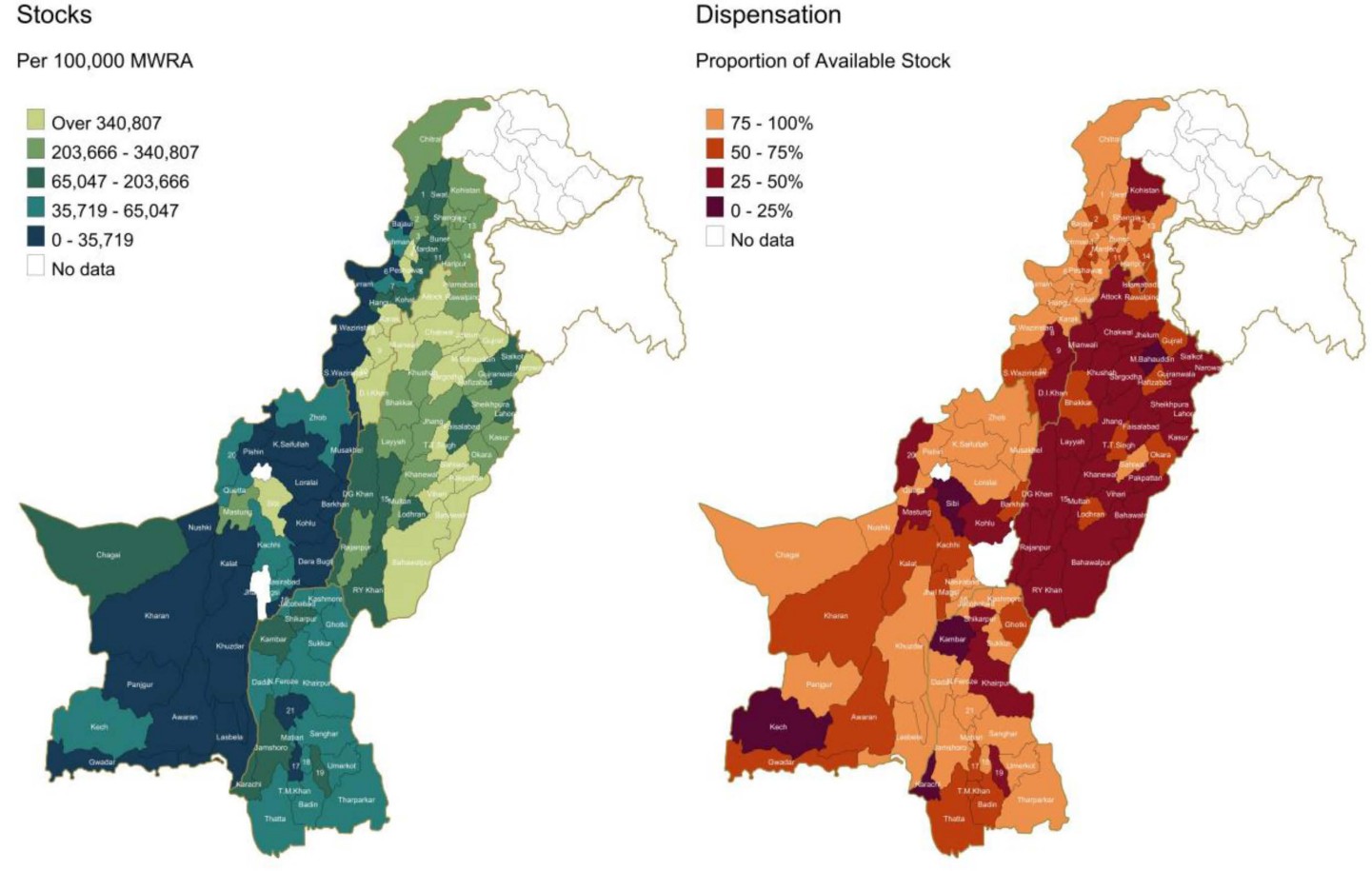

**Stocks**

Per 100,000 MWRA

- Over 340,807
- 203,666 – 340,807
- 65,047 – 203,666
- 35,719 – 65,047
- 0 – 35,719
- No data

**Dispensation**

Proportion of Available Stock

- 75 – 100%
- 50 – 75%
- 25 – 50%
- 0 – 25%
- No data

1.Upper Dir 2.Lower Dir 3.Malakand 4.Charsadda 5.Nowshera 6.Khyber 7.Orakzai 8.Bannu 9.Lakki Marwat 10.Tank 11.Swabi 12.Battagram 13.Mansehra 14.Abbottabad 15.Muzaffargarh 16.Jafarabad 17.Hyderabad 18.Tando Allahyar 19.Mirpurkhas 20.Killa Abdullah 21.Shaheed BenazirAbad

**Fig 2. Stocks and Dispensation of Condoms Adjusted for Population.** *Map Shape Files Source: Reprinted from* geoBoundaries *under a CC BY license, some modifications were made (included disputed area of Jammu & Kashmir) and used for illustrative purposes only.*

For injection stocks, Low–Low clusters were mostly in Balochistan, with a few High–High clusters in KP; dispensation Low–Low clusters were concentrated in Punjab and High–High clusters in central Sindh (Khairpur, Sanghar. Bivariate maps showed mixed Low–High and High–Low outliers, pointing to localized mismatches between supply and use. For IUD stocks and dispensation, Low–Low clusters were again primarily in Balochistan, with few High–High clusters elsewhere. Bivariate results showed minimal significant clustering, but some Low–High and High–Low patterns persisted.

Across commodities, Low–Low stock clusters reflect underserved regions—especially southern Balochistan—while Low–Low dispensation clusters occur mostly in central and southern Punjab. High–Low and Low–High outliers identify geographic mismatches between supply and uptake, highlighting priority areas for targeted interventions.

Maps for the spatial autocorrelation analyses are provided in S2 Appendix.

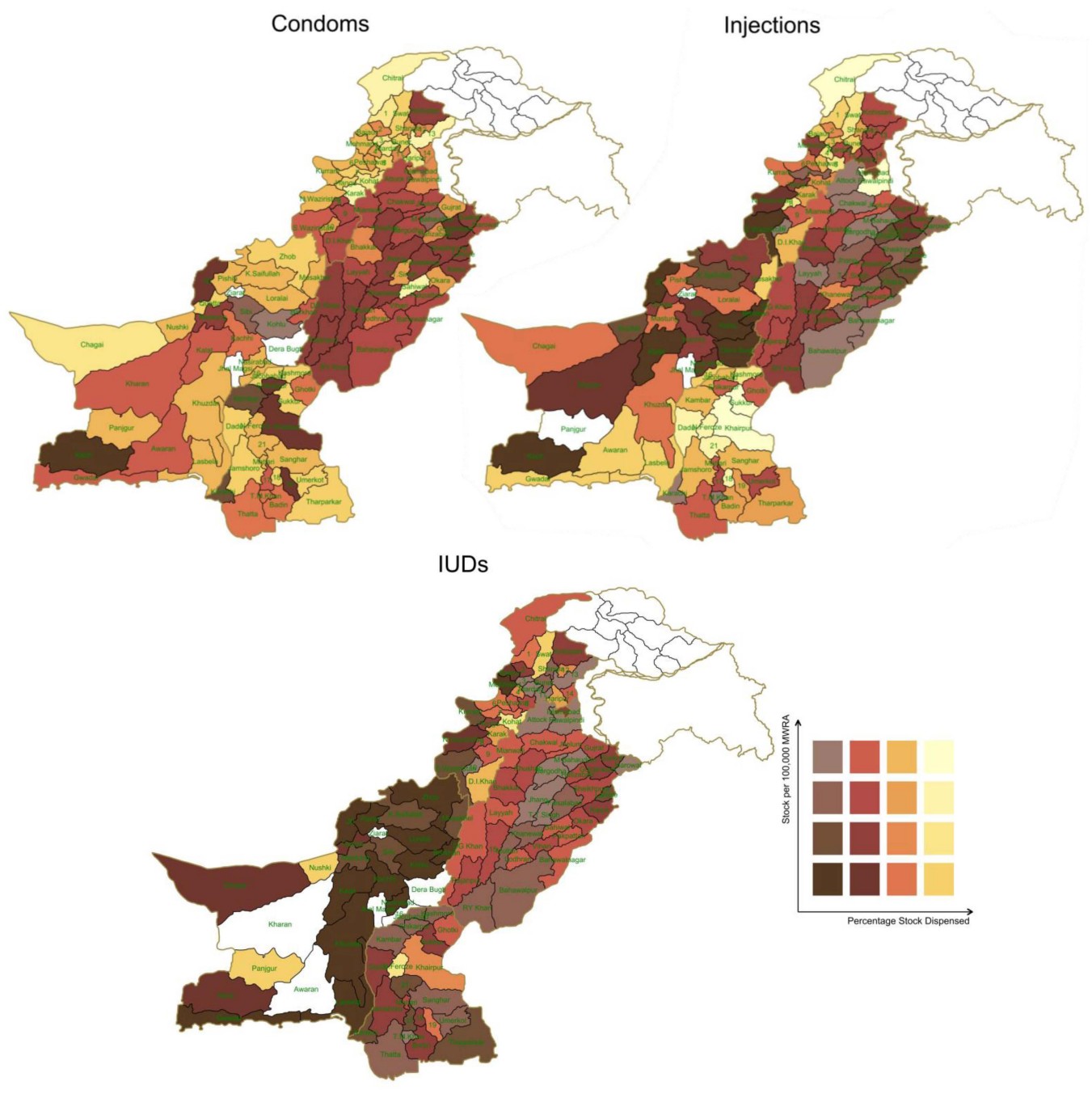

**Fig 3. Regional Disparities in Contraceptive Availability and Dispensation in Pakistan.** *Map Shape Files Source: Reprinted from* geoBoundaries *under a CC BY license, some modifications were made (included disputed area of Jammu & Kashmir) and used for illustrative purposes only.*

**Table 2. Local Moran's I Statistics and LISA Cluster Counts for Univariate and Bivariate Spatial Autocorrelation.**

| Commodity and *Indicator* | Moran I's Coefficient | Number (counts) of | | | |
|---|---|---|---|---|---|
| | | Low-Low Districts | Low-High Districts | High-Low Districts | High-High Districts |
| Condoms | | | | | |
| *Stock* | 0.179 | 26 | 3 | 0 | 4 |
| *Dispensation* | 0.038 | 17 | 3 | 1 | 2 |
| *Stock vs Dispensation* | −0.055 | 14 | 3 | 4 | 0 |
| *Dispensation vs Stock* | −0.057 | 12 | 6 | 11 | 1 |
| Injections | | | | | |
| *Stock* | 0.159 | 21 | 2 | 0 | 3 |
| *Dispensation* | 0.162 | 11 | 5 | 3 | 5 |
| *Stock vs Dispensation* | −0.016 | 11 | 7 | 3 | 2 |
| *Dispensation vs Stock* | −0.004 | 14 | 4 | 6 | 1 |
| IUDs | | | | | |
| *Stock* | 0.212 | 24 | 2 | 0 | 3 |
| *Dispensation* | 0.069 | 10 | 1 | 0 | 2 |
| *Stock vs Dispensation* | 0.019 | 9 | 3 | 0 | 0 |
| *Dispensation vs Stock* | 0.021 | 16 | 3 | 4 | 2 |

*Note: Low–Low clusters indicate districts with low values surrounded by neighbors with similarly low values. High–Low clusters indicate districts with high values surrounded by neighbors with low values. Low–High clusters indicate districts with low values surrounded by neighbors with high values. High–High clusters indicate districts with high values surrounded by neighbors with similarly high values.*

## Spatial and temporal insights using bivariate maps (immunization indicators as example)

Another layer of information can be incorporated to inform about the effect of time. Fig 4 shows monthly changes in the rate of dropouts for the Penta-3 vaccine as a factor of community outreach across various districts of the Sindh province. There are relatively high dropout rates in many southern and central districts in the first quarter, with some improvements by March. Tharparkar, Dadu and Khairpur consistently show high dropout rates throughout the quarter. Sukkur and Ghotki (in the north) show gradual improvements with relatively lower dropout rates and increasing outreach by the end of the quarter, particularly in March. These patterns can inform about seasonal variations or changes in the level of implementation of interventions over time.

## Deeper insights using three indicators in trivariate spatial map

In addition to the temporal changes the trivariate spatial map also allows adding a third dimension. For, e.g., Sindh province data is displayed for vaccine stock and dispensation (both per 1,000 targeted population) vs. the wastage shown as proportion of available stock across districts (Fig 5).

Fig 5 shows that southern (Korangi, Malir, Hyderabad and Matiari) and central (Jacobabad) districts have the highest stock levels per 1,000 targeted population, while some northern districts (Naushahro Feroze, Shikarpur, Ghotki and Shaheed Benazir Abad) have lower stock levels. Overall, adjusted stock levels show that there are ample stocks available per district. However, vaccination and wastage rates differ across districts.

High vaccination rates are observed in northern and central districts of Sindh such as Jacobabad, Dadu, Kashmore, Shaheed Benazir Abad, etc., whereas low vaccination rates in the south (Thatta, Hyderabad and all districts of Karachi), highlighting potential issues in either vaccine distribution or uptake. Significant vaccine wastage is observed, above the 50% threshold, in southern districts such as Thatta, Hyderabad, and all districts of Karachi, indicate inefficiencies in vaccine management. Lower vaccine wastage was observed in central and northern districts (Shaheed Benazir Abad,

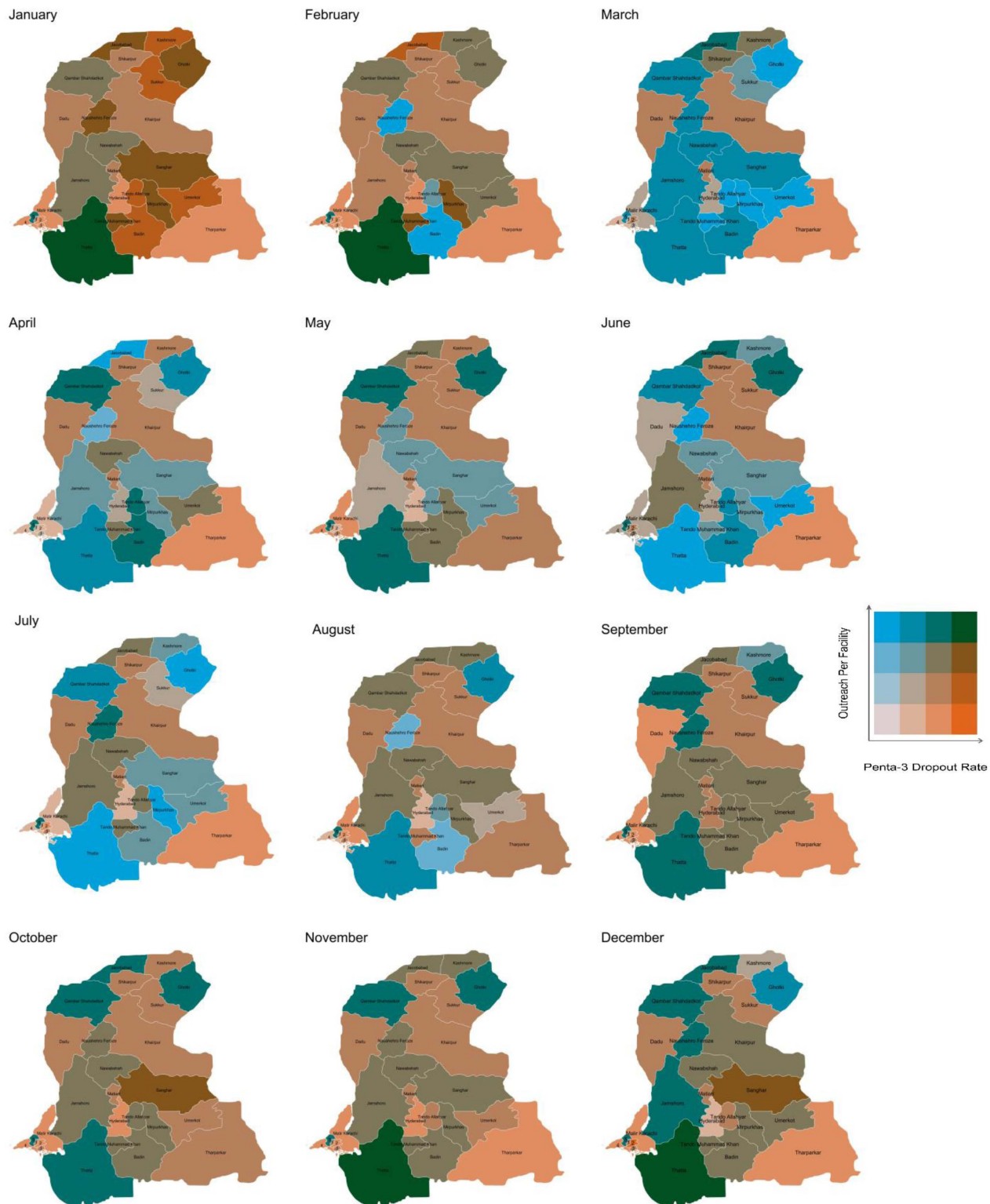

**Fig 4. Spatial and Temporal Variations in Outreach per Facility and Penta-3 Dropout Rates in Sindh.** *Note: For the districts marked with numbers: 1 indicates "Central Karachi", 2 indicates "East Karachi, 3 indicates "Korangi Karachi", 4 indicates "South Karachi" and 5 indicates "West Karachi. Map Shape Files Source: Reprinted from* geoBoundaries *under a CC BY license, some modifications were made (added six districts of Karachi) and used for illustrative purposes only.*

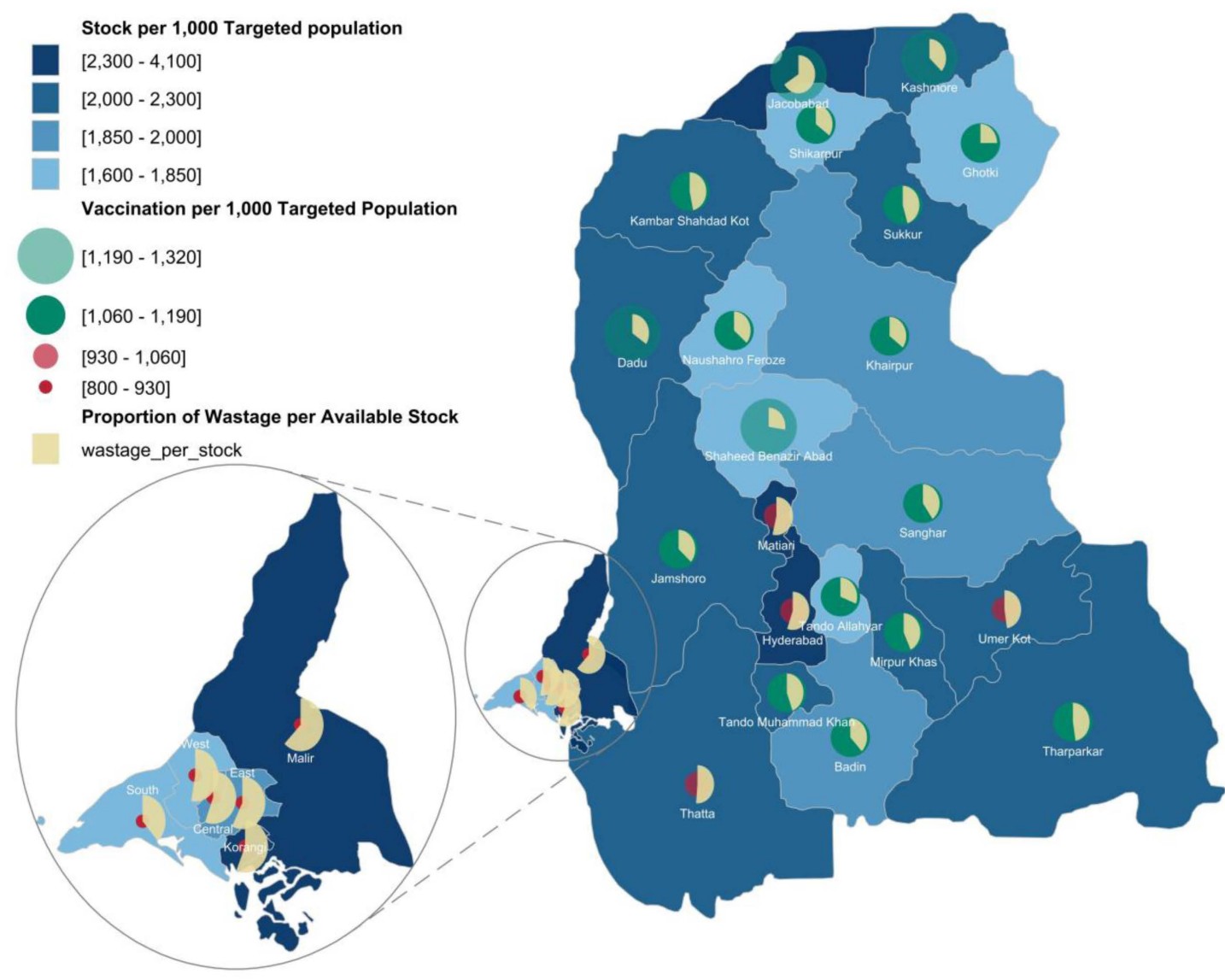

**Fig 5. Geospatial Mapping of BCG Vaccine Distribution, Uptake, and Wastage in Sindh.** Notes: Zoomed part (left of part of the Figure, below Legend) are the districts of Karachi City: South Karachi, West Karachi, Central Karachi, East Karachi, Korangi and Malir. Map Shape Files Source: Reprinted from geoBoundaries under a CC BY license, some modifications were made (added six districts of Karachi) and used for illustrative purposes only.

Naushahro Feroze, Shikarpur, and Ghotki). However, this must be interpreted along with the prior observation of low stocks, suggesting that these lower stocks are being dispensed.

## Discussion

Our research's spatial heat maps reveal significant regional disparities in the availability and dispensation of essential resources across Pakistan, particularly in contraceptives and immunization supplies. This highlights district-level supply and demand issues through specific indicators and underscores the need for resource redistribution or targeted

interventions across districts with higher stock levels and lower demand. For instance, if one district has surplus stocks, these can be redirected to neighboring districts facing shortages. This integration of data helps decision-makers understand the underlying problems better and reduces cognitive complexity by focusing on key factors, thus streamlining the decision-making process.

Furthermore, temporal-spatial heat maps enables the examination of variations over time. In Tharparkar district (Fig 4), outreach services remain consistently lower and Penta3 dropout rates are relatively high across all months, particularly when outreach declines. Analyzing spatial and temporal variations of the indicator enables us to track within or across districts, while risk maps based on the latest data enhance intervention measures [30]. Fig 5 highlights vaccine wastage as a major supply issue, analyzed using trivariate spatial heat maps. Addressing this requires reducing distribution inefficiencies in high-wastage, low-coverage districts, improving storage in high-wastage areas, and strengthening outreach and education in low-coverage regions to increase uptake.

Our spatial autocorrelation analysis provides the geographic context for these findings, confirming that the observed disparities are not random but follow clear spatial patterns of clustering and outliers. This strengthens the evidence base for targeted, location-specific interventions. For condom stocks, Low–Low clusters in southern Balochistan and much of Sindh indicate chronically underserved areas. Dispensation Low–Low clusters in central and southern Punjab suggest persistent access or demand constraints. Bivariate analysis showed High–Low clusters in parts of Balochistan and Sindh, pointing to stockout risks in high-demand areas, and Low–High clusters indicating inefficiencies in redistribution. Similar patterns emerged for injectables, with Low–Low stock clusters concentrated in Balochistan and mixed outliers signaling localized mismatches. IUDs patterns showed fewer significant clusters but persistent Low–Low stocks in Balochistan.

Low–Low clusters consistently mark areas where both supply and uptake lag, while High–Low and Low–High outliers highlight operational inefficiencies. Addressing these gaps calls for targeted redistribution in Low–Low regions, and stronger real-time monitoring to align supply with local demand across regions. Through monthly monitoring of these indicators, improvements can be achieved from within the existing system without requiring major new investments.

Exploratory data techniques effectively support multicriteria spatial decision-making by enhancing the interaction between maps and data graphs/points [31]. The integration of spatial mapping and analysis techniques has enabled the use of an expanding array of routinely available geographic data sources for public health care, research and development [32]. Spatial methods in epidemiology clarify the distribution of population health and its local causes, guiding targeted interventions to prevent disease and promote health [33]. Additionally, recent studies show that spatial heat maps help identify the prevalence and spread of infectious diseases, such as malaria and COVID-19 [30,34,35].

These visualization techniques provide powerful tools for policymakers, featuring user-friendly interfaces that require minimal technical expertise to generate specific visualizations. By offering detailed, district-level insights into stock levels, dispensation, and wastage, these visualizations facilitate data-driven decision-making, ultimately improving program efficiency and resource allocation to meet regional needs. Their scalability ensures a consistent framework for ongoing analysis and monitoring across different districts and overtime.

These spatial heat maps can serve as actionable decision-support tools for district and provincial health managers.

Once decision-makers or provincial health managers identify the concept and analysis syntax, they can embed analysis leading to such spatial heat maps directly into the database or data dashboards (such as DHIS2 or LMIS) using short programs called Application Programming Interfaces (APIs) as a next step [36]. Furthermore, interactive maps can significantly enhance spatial decision-making [37] upon clicking the district decision-maker can get necessary statistics of that particular district. When integrated into existing platforms dashboards, this facilitates rapid identification of underserved or over-served districts, timely redistribution of commodities, targeted outreach to high-dropout or zero-dose areas, and interventions to reduce wastage through strengthened storage and cold chain practices. To fully operationalize these tools, capacity building through short courses in GIS interpretation and spatial decision-making will be essential but these tools can be learned very quickly.

Such tools facilitate analysis at both the macro and micro levels, helping decision-makers identify overstocked or under-stocked locations, understand demand patterns, and identify resources for redistribution between neighboring districts or facilities, leading to informed decisions. Our findings contribute to visualizations that allow users to query large datasets with minimal human expertise, enabling better resource allocation at health facilities and ultimately improving health outcomes.

## Limitations

This study has several limitations. First, it relies on secondary data from the cLMIS and vLMIS systems, which may contain reporting inconsistencies or delays. This might overstate or understate some districts at one given time. Second, while heat maps are effective for visual interpretation, they do not provide causal explanations. Instead, they serve as diagnostic tools to flag anomalies for further investigation. Third, the study focused on variables available through national data systems rather than internationally standardized indicators. The analysis prioritized demand and supply metrics relevant to routine policy decisions, while interpreting demand as commodity dispensed in case of contraceptives. Demand represented in the research, is not equivalent to actual demand of using the contraceptive, rather it highlights the dispensed commodities which during interpretation can be overestimated compared to actual demand of using contraceptives. Fourth, interactive visualization was beyond the scope of this work; the intent was to illustrate that even such visual analytics can initiate data-informed decision-making and should be integrated into routine health system processes. Fifth, the study relies on population census data from 2017, which is somewhat outdated. The cutoff points are not intended to serve as reference benchmarks; rather, they are presented solely to illustrate disparities across regions.

## Conclusion

Our analysis of contraceptive and immunization data using bivariate and trivariate spatial heat maps reveals significant regional disparities in Pakistan. Urban centers, particularly in developed areas like Punjab and Sindh, show better stock availability than less developed regions such as Balochistan and southern Khyber Pakhtunkhwa. In immunization, our temporal and spatial analysis of Penta3 indicators highlights seasonal variations and persistent outreach challenges in many parts of Sindh province. This necessitates focused and resource-intensive strategies to improve vaccination coverage. Addressing vaccine wastage through improved distribution and storage, along with enhanced community outreach, is crucial.

Integrating data point and spatial maps enhances decision-making processes. These visualizations offer detailed insights into resource distribution, facilitating data-driven decisions that optimize program efficiency. Spatial maps simplify the generation of actionable insights, empowering policymakers to make informed decisions quickly. Ultimately, spatial maps serve as essential tools for guiding health interventions, ensuring effective resource allocation to meet diverse regional needs. This research highlights the critical role of spatial analysis in improving public health outcomes through targeted policy actions.

## Supporting information

**S1 Appendix. Spatial Autocorrelations for Contraceptives.**
(DOCX)

**S2 Appendix. Cluster and Significance Maps for Univariate and Bivariate Spatial Autocorrelation Maps.**
(DOCX)

## Acknowledgments

We would like to extend our gratitude to CHEMONICS Pakistan for providing the valuable data used in this study. Their contribution has been instrumental in enabling our research and enhancing our understanding of the supply and dispensation dynamics of contraceptives in Pakistan.

## Author contributions

**Conceptualization:** Muhammad Ibrahim, Olan Naz, Adnan Ahmad Khan.

**Data curation:** Habib Ur Rehman.

**Formal analysis:** Muhammad Ibrahim, Olan Naz, Amal Fatima Mohiuddin.

**Funding acquisition:** Adnan Ahmad Khan.

**Methodology:** Muhammad Ibrahim.

**Project administration:** Olan Naz, Amal Fatima Mohiuddin.

**Resources:** Adnan Ahmad Khan.

**Software:** Muhammad Ibrahim, Olan Naz, Habib Ur Rehman.

**Supervision:** Muhammad Ibrahim.

**Validation:** Adnan Ahmad Khan.

**Visualization:** Muhammad Ibrahim, Olan Naz, Amal Fatima Mohiuddin.

**Writing – original draft:** Muhammad Ibrahim, Olan Naz, Habib Ur Rehman.

**Writing – review & editing:** Muhammad Ibrahim, Amal Fatima Mohiuddin, Adnan Ahmad Khan.

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
