## [Decision Letter · Decision Letter 0]

12 Apr 2025

Dear Dr. Khan,

We look forward to receiving your revised manuscript.

Kind regards,

Syed Khurram Azmat, PhD, MPH, MD

Academic Editor

PLOS ONE

Journal Requirements:

2. In the online submission form, you indicated that [Insert text from online submission form here].

3. In the online submission form, you indicated that your data is available only on request from a third party. Please note that your Data Availability Statement is currently missing [contact details for the third party, such as an email address or a link to where data requests can be made]. Please update your statement with the missing information.

4. Please amend your manuscript to include your abstract after the title page.

5. We note that Figures 1-5 in your submission contain [map/satellite] images which may be copyrighted. All PLOS content is published under the Creative Commons Attribution License (CC BY 4.0), which means that the manuscript, images, and Supporting Information files will be freely available online, and any third party is permitted to access, download, copy, distribute, and use these materials in any way, even commercially, with proper attribution. For these reasons, we cannot publish previously copyrighted maps or satellite images created using proprietary data, such as Google software (Google Maps, Street View, and Earth). For more information, see our copyright guidelines: http://journals.plos.org/plosone/s/licenses-and-copyright.

a. You may seek permission from the original copyright holder of Figures 1-5 to publish the content specifically under the CC BY 4.0 license. 

Reviewers' comments:

Reviewer's Responses to Questions

**Comments to the Author**

1. Is the manuscript technically sound, and do the data support the conclusions?

Reviewer #1: Yes

Reviewer #2: Partly

Reviewer #3: Partly

2. Has the statistical analysis been performed appropriately and rigorously?

Reviewer #1: Yes

Reviewer #2: Yes

Reviewer #3: Yes

3. Have the authors made all data underlying the findings in their manuscript fully available?

Reviewer #1: No

Reviewer #2: Yes

Reviewer #3: No

4. Is the manuscript presented in an intelligible fashion and written in standard English?

Reviewer #1: Yes

Reviewer #2: Yes

Reviewer #3: No

Reviewer #1: Dear Authors,

Congratulations on preparing a well-structured and logically coherent manuscript. The document is well written; however, some revisions are necessary to enhance its clarity and completeness. The reviewer's suggestions and comments are noted in the attached document for your consideration.

It appears that the final manuscript was not thoroughly reviewed before submission, and certain sections would benefit from further elaboration and/or clarification. Additionally, the references do not adhere to a standard format and should be revised accordingly.

I am recommending minor revisions, after which the manuscript can be resubmitted for publication. Wishing the authors the best of luck in the revision process.

Best regards,

Reviewer #2: Major comment

The manuscript introduces an interesting approach towards analysing contraceptive availability and consumption data along with vaccine availability and administration in Pakistan. This approach per se is not novel, however, provides some level of insight into these issues in the context of Pakistan. The manuscript does not fully justify how and why the performed analysis may be superior to techniques already deployed and in use.

Abstract

The authors mention the effect of inputs on “rate of utilization of services”. The paper does not mention any rate(s) either for contraceptive use or for immunization.

The objective appears to be “We explore the use of spatial heat maps to analyze the distribution of contraceptives and vaccines in Pakistan and to depict these analyses as visualizations.” However, the data from cLMIS is merely the stock of contraceptive(s), not distribution. How do the authors plan to address this aspect?

The results section states “we also show the effect of inputs (supplies, outreach) on rate of utilization of services (contraceptive uptake and vaccine coverage). Finally we depict how these visualizations can help track changes in programming over time.” This is merely a claim and not the finding or result of the study. For example, what is the “rate of utilization of services” the authors have measured? What are the findings that help track change over time?

The authors conclude that they the study “identifies critical gaps in health service supply and demand”, but it is not clear what is the gap?

Specific comments

There is a question of generalizability of this study. The authors have referred to its application to all LMICs in the conclusion. However, does Pakistan reflect all LMICs, more so since vaccination data analysis covers the province of Sindh and that too for BCG and Penta-3 via outreach facilities only?

Please revisit the introduction section for more clarity. There are various language errors, note lines 35, 39, 48, 53, 59, 61, 69 and others.

Lines 74-78 appear to be more of a discussion point rather than introduction.

Objective is not clear in the introduction section.

More information on facilities which report data both for cLMIS and vLMIS will be helpful. Are these facilities all in the public sector or in the private sector as well. For contraceptive data, are the facilities operated by the health departments or the population departments in respective provinces. Is contraceptive availability captured in from the private sector as well. Does only outreach Penta-3 coverage provide sufficient information for this kind on analysis?

For contraceptives the authors have stated that the consumption data does not mean that contraceptives have been utilized (line 97). In this situation how can it be suggested that rates of utilization of services have been measured (see abstract).

It is unclear from the analysis if private sector information is part of this data set. Implications will be different if it is than if it is not. Please ensure that it is made clear.

Discussion section needs to be revisited. It will benefit from a discussion around the findings, that is how does the spatial analysis report the status of the contraceptive and vaccine availability as well as consumption. What are the regional disparities? A discussion of trends.

Please also discuss potential limitations in some detail.

The authors state that, “policymakers should address distribution inefficiencies in districts with high wastage and low vaccination rates, improve storage facilities in areas with high wastage, and enhance community outreach and educational campaigns in regions with low vaccination rates to boost uptake.” How is this recommendation flow from the spatial analysis conducted in this study? Isn’t this already known, BCG has one of the highest wastage rates among vaccines.

What policy level decisions can be taken by policy makers using this approach of data analysis?

The discussion section talks about some dashboard and policy makers clicking to get statistics of a district. Was that the purpose of this study? It is not relevant and confuses the reader.

Discussion needs to be crisp and avoid repetition which is considerable at present.

The manuscript does not convince the reader how the use of visualizations may improve routine data analysis and interpretation in this context.

Reviewer #3: It is an immensely useful topic to investigate for country with massive population growth and sub-optimal policy development process.

Major areas of improvement for this mansucript are language and grammatical errors, logical flow and failure to acknowledge the practical and methodological limitations.

'Consumption' is a key variable and intricate details are lacking, how it is calcuated.

**Do you want your identity to be public for this peer review?** For information about this choice, including consent withdrawal, please see our Privacy Policy

Reviewer #1: **Yes: ** SYED FARHAN ALI TIRMIZI

Reviewer #2: No

Reviewer #3: **Yes: ** Fahad Javaid Siddiqui

---

## [Author Response · Author response to Decision Letter 1]

30 May 2025

We thank the reviewers for thorough review of our manuscript and acknowledging the importance of the topic we have done our research for. Responses to the comments have been given in the word file shared on the portal.

---

## [Decision Letter · Decision Letter 1]

4 Aug 2025

Dear Dr. Khan,

The reviewers and editorial team acknowledge that you have addressed the majority of the concerns raised during the previous round of review. Your responses have significantly improved the manuscript, and we commend your efforts to enhance the clarity and rigor of the work. However, there are still a few remaining areas that require attention before the manuscript can be accepted for publication (refer to the second peer review report/s).

We therefore invite you to submit a minor revision of your manuscript. Please ensure that you provide a detailed, point-by-point response to the remaining concerns, highlighting any changes made in the revised manuscript.

We look forward to receiving your revised manuscript.

Kind regards,

Syed Khurram Azmat, PhD, MPH, MD

Academic Editor

PLOS ONE

**Journal Requirements:**

Reviewers' comments:

Reviewer's Responses to Questions

**Comments to the Author**

Reviewer #2: (No Response)

Reviewer #4: (No Response)

2. Is the manuscript technically sound, and do the data support the conclusions?

Reviewer #2: Partly

Reviewer #4: (No Response)

3. Has the statistical analysis been performed appropriately and rigorously?

Reviewer #2: Yes

Reviewer #4: (No Response)

4. Have the authors made all data underlying the findings in their manuscript fully available?

Reviewer #2: Yes

Reviewer #4: (No Response)

5. Is the manuscript presented in an intelligible fashion and written in standard English?

Reviewer #2: Yes

Reviewer #4: (No Response)

**Reviewer #2: ** Two concerns still remain.

1. Reporting on 'consumption'. While the authors have tried to address this by adding a definition for consumption, by their own admission "consumption alone does not indicate whether the available stocks are being appropriately utilized" (178-179). This contradicts with their assertion that "the proportion of stocks consumed serves as an indicator of demand for available

commodities. For example, if consumption falls below 25% or 50% of the available stock, it suggests under utilization and potential misalignment between supply and demand" (208-210).

The fundamental point is that cLMIS data does not provide consumption information for contraceptives such as condoms and oral pills. This is only 'issuance' or 'dispensation' data. Once a client is given these contraceptives, only clients can provide the information on whether the method has been utilized or otherwise. This is a serious limitation in this analysis and has not been discussed in the relevant section.

2. Reporting 'trends'. The data represented from a single year i.e. 2022. The only trends provided are for Penta-3 vaccine and that too only for 12 months in a single year. The paper is presenting this as a trend analysis for contraceptives as well, which is not correct.

These two aspects must be rectified.

**Reviewer #4: ** Reviewer Report for PONE-D-25-06311 / PONE-D-25-06311R1

1. Previous Round Response

The authors have provided a “Response to Reviewers” file indicating that they revised the manuscript throughout. Based on my review below, I do not recommend bypassing detailed comments; the manuscript still requires some improvements before acceptance.

2. Technical Soundness & Data Support

Recommendation: Minor revision

The study applies heat‐map visualizations coherently and the data presented support the stated conclusions about geographic disparities and temporal trends in contraceptive and vaccine logistics. However, the manuscript lacks a more rigorous description of any statistical testing or validation of the quantile cut‐offs and does not discuss uncertainty or sensitivity of the maps.

3. Statistical Analysis

Recommendation: Needs improvement

While the mapping methods are clearly described, no statistical tests (e.g., spatial autocorrelation measures, significance of differences between regions, or confidence intervals for consumption proportions) are reported. The choice of quartile cut‐offs should be justified or tested for robustness.

4. Data Availability

Recommendation: No

The Data Availability Statement indicates that data “will be available upon request” from Chemonics, which does not satisfy PLOS ONE’s requirement for fully open data. The authors must deposit cleaned, de‐identified datasets and any shapefiles to a public repository (e.g., Dryad, Zenodo) and include accession DOIs.

5. English & Presentation

Recommendation: Acceptable with minor editing

Overall the English is clear and the flow logical. A few grammatical or typographical errors remain (e.g., “adjected” → “adjusted” in Figure 2 caption; inconsistent hyphenation of “bi-variate” vs. “bivariate”). The Introduction and Discussion could be tightened to avoid some repetition.

6. Review Comments to the Author

Abstract & Keywords

Change made: Abstract wording refined for clarity; “depict these analyses as visualizations” changed to “analyze the stock availability and consumption/vaccination patterns” and keywords expanded to include “public policy” and “immunization.”

Suggestion: Remove redundancies (“Finally, we depict…” duplicates methods wording). Limit keywords to six.

Methods – Data Sources & Shapefiles

Change made: Added description of geoBoundaries shapefiles and CC BY 4.0 license.

Suggestion: Move shapefile license information to a Data Availability subsection. Clarify any QA/QC steps taken on LMIS data (e.g., how missing or delayed reports were handled).

Statistical Rigor

Needs improvement: No statistical measures accompany the maps. Consider adding:

Spatial autocorrelation (e.g., Moran’s I) to quantify clustering.

Sensitivity analysis around quartile cut-offs.

Confidence intervals or bootstrapping for consumption proportions.

Data Availability

Needs improvement: PLOS policy requires public deposition. Please upload aggregated, de-identified district-level data and shapefiles to a public repository (e.g., Zenodo) and include DOI in the Data Availability Statement.

Ethics Statement

Current: “N/A”

Suggestion: Although this is secondary data, state explicitly “Data are aggregate and anonymized; ethics approval was not required” or cite the institutional review board that waived review.

English & Formatting

Minor corrections:

Figure 2 caption: “adjected” → “adjusted.”

Consistency in hyphenation: use “bivariate” and “trivariate” throughout.

Remove duplicate paragraphs in Discussion beginning “Improving data systems…” and “Exploratory data techniques…”

Limitations

Needs improvement: Although briefly mentioned, the Limitations section should discuss potential biases due to delayed reporting, data entry errors, and the exclusion of private sector LMIS gaps. Also note that MWRA estimates are based on the 2017 census only.

Figures & Legends

Change made: Figures reorganized and legends refined.

Suggestion: Ensure all maps have consistent color scales and that size/area legends (e.g., circle sizes in tri-variate maps) are clearly labeled with numeric values.

Discussion and Conclusions

Needs improvement: The Discussion reiterates results without deeper interpretation of policy implications. Please expand on how local health managers could operationalize these heat maps (e.g., integrating into existing dashboards, training needs).

**Do you want your identity to be public for this peer review?** For information about this choice, including consent withdrawal, please see our Privacy Policy

Reviewer #2: No

Reviewer #4: **Yes: ** Muhammad Bilal Siddiqui

---

## [Author Response · Author response to Decision Letter 2]

15 Aug 2025

We thank the reviewers for the thorough review of our manuscript and highlighting the errors and suggesting recommendations to make our manuscript better for the readers. Especially, thank reviewer 2 for highlighting using terms that can be misleading to the readers and then thanks for reviewer 4 for highlighting the use of statistical analysis in the our manuscript.

---

## [Editor Report · Decision Letter 2]

1 Sep 2025

Unlocking Insights from Complex Data: Leveraging Heat Maps for Decision-Making in LMIC

PONE-D-25-06311R2

Dear Dr. Khan,

We’re pleased to inform you that your manuscript has been judged scientifically suitable for publication and will be formally accepted for publication once it meets all outstanding technical requirements.

Kind regards,

Syed Khurram Azmat, PhD, MPH, MD

Academic Editor

PLOS ONE
---

## [Editor Report · Acceptance letter]

PONE-D-25-06311R2

PLOS ONE

Dear Dr. Khan,

I'm pleased to inform you that your manuscript has been deemed suitable for publication in PLOS ONE. Congratulations! Your manuscript is now being handed over to our production team.

Kind regards,

on behalf of

Dr. Syed Khurram Azmat

Academic Editor

PLOS ONE